# Comparative Efficacy of Parenteral and Mucosal Recombinant Probiotic Vaccines Against SARS-CoV-2 and *S. pneumoniae* Infections in Animal Models

**DOI:** 10.3390/vaccines12101195

**Published:** 2024-10-19

**Authors:** Galina Leontieva, Tatiana Kramskaya, Tatiana Gupalova, Elena Bormotova, Yulia Desheva, Dmitry Korzhevsky, Olga Kirik, Irina Koroleva, Sergey Borisevitch, Alexander Suvorov

**Affiliations:** 1Scientific and Educational Center “Molecular Bases of Interaction of Microorganisms and Human” of the World-Class Research Center “Center for Personalized Medicine”, Federal State Budgetary Scientific Institution «Institute of Experimental Medicine» (FSBSI «IEM»), 197376 Saint Petersburg, Russia; kramskaja.ta@iemspb.ru (T.K.); tvgupalova@rambler.ru (T.G.); bormotovae@rambler.ru (E.B.); desheva@mail.ru (Y.D.); olga_kirik@mail.ru (O.K.); suvorov.an@iemspb.ru (A.S.); 2Federal State Budgetary Science Institute “IEM”, 197376 Saint Petersburg, Russia; dek2@yandex.ru; 3Federal State Budgetary Institution 48th Central Research Institute of the Ministry of Defense of the Russian Federation, 141306 Moscow, Russia; 48cnii@mil.ru

**Keywords:** recombinant-probiotic-based vaccines, probiotic strain *E. faecium* L3, mucosal vaccines, parenteral vaccines, oral vaccination, vaccine efficacy

## Abstract

Background: The accumulation of specific IgG antibodies in blood serum is considered a key criterion for the effectiveness of vaccination. For several vaccine-preventable infections, quantitative indicators of the humoral response have been established, which, when reached, provide a high probability of protection against infection. The presence of such a formal correlate of vaccine effectiveness is crucial, for example, in organizing preventive measures and validating newly developed vaccines. However, can effective protection against infection occur when the level of serum antibodies is lower than that provided by parenteral vaccination? Will protection be sufficient if the same vaccine antigen is administered via mucosal membranes without achieving high levels of specific IgG circulating in the blood? Methods: In this study, we compared the immunogenicity and protective efficacy of parenteral and mucosal forms of vaccines in experimental animals, targeting infections caused by the SARS-CoV-2 coronavirus and *Streptococcus pneumoniae*. We investigated the protective properties of a fragment of the coronavirus S1 protein administered intramuscularly with an adjuvant and orally as part of the probiotic strain *Enterococcus faecium* L3 in a Syrian hamster model. A comparative assessment of the immunogenicity and protective efficacy of a recombinant tandem (PSP) of immunogenic peptides from *S. pneumoniae* surface proteins, administered either parenterally or orally, was performed in a Balb/c mouse model. Results: Both models demonstrated significant differences in the immunogenicity of parenteral and oral vaccine antigens, but comparable protective efficacy.

## 1. Introduction

Vaccination has long been, and continues to be, the most important and effective method of protection against infectious diseases [1]. The invention of the hypodermic syringe in the 1850s [2] enabled controlled vaccine delivery and expanded the practice of parenteral vaccination, making it a cornerstone of public health [3]. Since then, numerous parenteral vaccines have been developed, targeting a wide range of infectious diseases and population groups. These include vaccines that have been instrumental in controlling several serious infections, such as smallpox, polio, measles, diphtheria, tetanus, hepatitis B, *Haemophilus influenzae* type b (Hib), human papillomavirus (HPV), and SARS-CoV-2.

Injectable vaccines provide both protective efficacy and the ability to assess it by measuring circulating IgM and IgG antibodies. However, most human pathogens enter the host not through the bloodstream but via the natural system of host defenses—mucosal surfaces. It is well known that the predominant class of antibodies at mucosal surfaces is IgA, which plays a critical role in controlling pathogenic bacteria and viruses at these primary sites of infection. Mucosal vaccination, administered orally, nasally, vaginally, or intranasally, offers a logical alternative to parenteral vaccination. However, the development of mucosal vaccines has been slow, and most licensed vaccines today remain injectable [4]. Notably, in the early 20th century, Russian scientist A. Bezredka demonstrated the high protective potential, safety, and ease of use of an oral cholera vaccine during cholera outbreaks in India [5]. Despite this early success, the oral cholera vaccine Dukoral was not licensed until 1991 [6].

The introduction of mucosal vaccines into clinical practice presents several scientific and technical challenges, one of the most important being the selection of an antigen delivery system that ensures the vaccine reaches mucosal surfaces in a form capable of eliciting an adequate immune response. One promising approach is the use of probiotic bacteria as antigen delivery vehicles. Through genetic modification, a DNA fragment encoding the desired vaccine antigen is introduced into the probiotic strain. The first studies of recombinant probiotic vaccines appeared in the scientific literature in the late 1990s and early 2000s [7]. These studies investigated the use of genetically modified probiotic bacteria, such as *Lactobacillus* and *Escherichia coli* strains, as vaccine antigen delivery systems. Over time, recombinant probiotics have been developed to express a wide range of antigens, including viral proteins, bacterial toxins, and cancer-associated antigens, with potential applications in both human and veterinary medicine [8,9].

In this context, we have developed and are currently evaluating recombinant vaccines based on the probiotic strain *Enterococcus faecium* L3. We have created enterococcal delivery vehicles expressing several bacterial and viral antigens [10,11,12,13]. Experimental models have demonstrated the efficacy of these vaccines in protecting against infections. We, along with others working on mucosal vaccines, have observed that the protective effect of mucosal vaccination often occurs in conjunction with a relatively weaker humoral immune response compared to parenteral vaccination. These results, along with data suggesting that a strong systemic immune response may not always be beneficial—and may sometimes be harmful, as in the case of antibody-dependent enhancement (ADE) [14]—lead us to believe that in some cases, oral vaccination could replace parenteral vaccination if the protective potential of both routes is comparable.

This study compares the efficacy of mucosal and parenteral vaccination in experimental animals using a probiotic-based mucosal vaccine and an injectable parenteral form of vaccine proteins. Specifically, we performed a comparative analysis of the immunogenicity of two vaccine antigens, one viral (SARS-CoV-2) and one bacterial (*Streptococcus pneumoniae*), following parenteral immunization with an adjuvant and oral administration using a recombinant probiotic vaccine. Using experimental models of coronavirus and pneumococcal infections, we evaluated the protective potential of the immune responses elicited by both vaccine formulations. The results demonstrated equivalent efficacy for both vaccination routes, using vaccine proteins of both viral and bacterial origin.

## 2. Materials and Methods

### 2.1. Cell Culture

Vero C1008 cells (ECACC: 85020206) were employed for in vitro experiments. The cells were cultured in Minimum Essential Medium (MEM) (PanAco, Ribinsk, Russia) with 2% FBS (Gemini, Sacramento, CA, USA) and grown in 225 cm^2^ flasks (Cellstar, Greiner Bio-One GmbH, Frickenhausen, Germany). Subculturing was performed every 3 to 4 days using trypsin to detach the monolayers.

### 2.2. Virus

The SARS-CoV-2 strain utilized in this study, hCoV-19/Russia/SAB-1502/2021, was sourced from the Federal Budgetary Research Institution-State Research Center of Virology and Biotechnology “VECTOR”. It was part of the South Africa/Gamma 1.351 501.V2 lineage, featuring the mutations S D80A, D215G, E484K, and N501Y.

#### 2.2.1. Assessment of Infectious Activity

The infectious activity of SARS-CoV-2 was assessed using two methods: a plaque assay in Vero cell monolayers under solid overlay media and a TCID_50_ assay in Vero cell monolayers [15].

#### 2.2.2. Preparation of Virus for Challenge

Prior to the challenge, the SARS-CoV-2 virus was propagated in Vero cell culture. A cell monolayer was established by seeding 2 × 10^5^ cells/mL in plastic flasks (Cellstar, Greiner Bio-One GmbH, Frickenhausen, Germany) and incubating for 24 h at 37.0 °C with 5% CO_2_. The growth medium consisted of modified Eagle’s medium (MEM, Gibco, Thermo Scientific, Waltham, MA USA) supplemented with 2% FBS (Gemini, Sacramento, CA, USA), 100 μg/mL penicillin, and 100 μg/mL streptomycin. Virus adsorption was performed by adding the virus at a concentration of 1 PFU/mL to the cell monolayer and incubating for 60 min at 37.0 °C. Following adsorption, the inoculum was removed, the cells were rinsed with MEM, and 7–8 mL of fresh growth medium was added. The flasks were then incubated at 37.0 °C with 5% CO_2_ for 48 h. After incubation, the cells were harvested, and the supernatant was separated by centrifugation, aliquoted, and stored at −70 °C. The virus preparation was assessed for sterility and infectious activity using plaque and TCID_50_ assays.

### 2.3. Bacteria

The *Enterococcus faecium* strain L3 and its two recombinant derivatives, L3-S1 (expressing the S1 gene fragment from SARS-CoV-2) and L3-PSP (expressing a chimeric protein corresponding to surface antigens of *Streptococcus pneumoniae*), were obtained from the collection of the Institute of Experimental Medicine. All strains were grown in Todd Hewitt Broth (THB) (HiMedia, Mumbai, India) at 37 °C with shaking for 14 h. For bacterial cultivation, quantification, and identification, LB agar (Thermo Fisher Scientific, Waltham, MA USA) and Enterococcus Differential Agar Base (TITG, HiMedia, Mumbai, India) were used, with or without 10 μg/mL erythromycin, including for erythromycin-resistant enterococcal transformants. A clinical isolate of *Streptococcus pneumoniae* serotype 3 strain 73 was cultured for 18 h at 37 °C in 5% CO_2_ in THB medium (HiMedia, Mumbai, India) supplemented with 20% horse serum (Difco, Carrickmore, UK). For solid medium cultivation and bacterial enumeration, Columbia agar containing 10% horse serum and 5% defibrinated sheep blood was employed.

### 2.4. Animals

Syrian golden hamsters (50–60 g) were obtained from the Andreevka Branch of the FSBSN NTBMT FMBA and housed in a barrier facility under controlled conditions with a 3-day acclimation period. The hamsters were fed a pelleted diet, housed in plastic cages with REHOFIX^®^ bedding, and kept at 15–21 °C with 30–70% humidity and a 12 h light/dark cycle. Female Balb/c mice (10 weeks old) were sourced from “Rappolovo” in the Leningrad Region, Russia, and housed under standard conditions with free access to food and water.

### 2.5. Animal Procedures

#### 2.5.1. Hamsters

Before immunization, Syrian golden hamsters were evaluated for health, weighed for grouping, and divided into four experimental groups (*n* = 20 per group), as shown in Figure 1A.

The recombinant probiotic vaccine candidate L3-S1 and the recipient strain *E. faecium* L3 were administered orally at 1 × 10^9^ CFU in 0.1 mL PBS, once daily for three days. A second round followed three weeks later. Recombinant S1 protein (40 μg) with aluminum hydroxide was administered intramuscularly twice, three weeks apart.

Twenty-eight days after the second immunization, blood and swab samples were collected from five animals per group to assess immunogenicity. Two days later, hamsters were orally challenged with SARS-CoV-2 (4.3 log PFU). On days 3 and 6 post-inoculation, lung samples from euthanized hamsters were collected for viral replication analysis and histological examination.

#### 2.5.2. Mice

##### Immunization

Before immunization, Balb/c mice were assessed for health and quality. The animals were weighed and divided into four experimental groups (*n* = 20/group). Additionally, six untreated mice were used as a source of serum and lungs for histological analysis. The experimental design is illustrated in Figure 1B.

Two experimental groups received either the live recombinant probiotic vaccine candidate L3-PSP or the recipient strain *E. faecium* L3. These were administered orally in a 0.1 mL PBS suspension at a dose of 1 × 10^8^ CFU. The first round of vaccination was conducted once daily for three consecutive days, followed by a second round three weeks later.

Additionally, the third group of mice received a subcutaneous administration of an aqueous solution containing 20 μg of the recombinant PSP protein per dose, mixed with aluminum hydroxide as an adjuvant (2:1 ratio), in a total volume of 200 μL. The injections were given twice, with a three-week interval between doses. The fourth group received the adjuvant alone, following the same schedule.

Fourteen days after the initial vaccination and 28 days after the second immunization, blood and swab samples (a mixture of saliva and nasal secretions) were collected from six animals per group to evaluate the immunogenicity of the vaccine candidates. Blood was drawn under anesthesia from the subclavian vein, and the animals were euthanized by cervical dislocation.

##### Evaluation of Active Immune Protection

On day 52 post-immunization, or one month after the booster dose, the mice received an intranasal inoculation of 25 μL of Streptococcus pneumoniae at a dose of 5.0 log_10_ CFU. Then, 24 and 48 h post-inoculation, five mice per group were euthanized by cervical dislocation, and lung samples were collected to assess bacterial burden. The tissues were homogenized in PBS, and the resulting 10% homogenates were analyzed on Columbia agar plates containing 10% horse serum and 5% defibrinated sheep blood to evaluate the *S. pneumoniae* count.

To assess survival, some mice (*n* = 5/group) were monitored for mortality, with rates recorded on day 10 following the onset of infection.

##### Evaluation of Passive Immune Protection

For evaluating passive immune protection against S. pneumoniae infection, serum pools were prepared by combining equal aliquots from vaccinated and control mice. A similar serum pool was prepared from six untreated mice. The pooled sera were diluted 5-fold and mixed in a 1:1 ratio with an *S. pneumoniae* suspension at a concentration of 1 × 10^8^ CFU/mL. The mixture was incubated at 37 °C for 30 min, and then 40 μL of the sample was administered intranasally to mice (*n* = 5). This resulted in an infectious dose of 4 × 10^5^ CFU. Lung samples were collected from the mice at 24 and 48 h post-inoculation to assess bacterial load.

### 2.6. ELISA Assay

The ELISA used trimeric WT S protein (Vector-Best, Koltsovo, Russia) and recombinant S1 and PSP proteins corresponding to L3-S1 and L3-PSP vaccine strains. The procedure was performed as described by Gupalova et al., 2019 [10]. Maxisorb 96-well plates (Nunc, Roskilde, Denmark) were coated overnight at 4 °C with 0.25 μg/mL S1 and PSP proteins in sodium carbonate buffer (pH 9.3). After adding serial dilutions of samples (100 μL), plates were incubated for 1 h at 37 °C. HRP-labeled goat anti-hamster IgA or IgG antibodies were added (100 μL/well), followed by 1 h of incubation at 37 °C by TMB substrate (BD Bio-science, Franklin Lakes, NJ, USA) for detection. The reaction was stopped with 30 μL of 50% sulfuric acid, and ELISA titers were determined as the highest dilution with an OD450 above the negative control threshold.

### 2.7. Evaluation of Virus Neutralizing Activity of Serum and Swabs

#### 2.7.1. Plaque Reduction Neutralization Test on Vero Cell Culture Monolayers

Neutralizing antibodies against 100 PFU/mL of SARS-CoV-2 were assessed using a plaque reduction neutralization test (PRNT) on Vero cell monolayers. Two-fold dilutions of heat-inactivated hamster sera were tested in quadruplicate, with cytopathic effects evaluated on day 4. Antibody titers were defined as the highest serum dilution showing more than 50% plaque reduction compared to the negative control.

#### 2.7.2. Neutralizing Properties via Inhibition of SARS-CoV-2 S1 Protein Binding to ACE2

Neutralizing activity was assessed by the ability to block SARS-CoV-2 S1 protein binding to human ACE2 in an ELISA assay, as was described in detail in [12]. In short, microtiter plates were coated with 100 μL of ACE2 (1.6 μg/mL, HyTest, Moscow, Russia) in PBS (pH 7.4) and incubated for 24 h at 4 °C. Serum samples and nasopharyngeal washes (diluted 1:16) were incubated for 15 min at 37 °C while shaking with HRP-conjugated SARS-CoV-2 S glycoprotein (150 μL, Wuhan-Hu-1, ID: 43740568), before being transferred to ACE2-coated wells. After incubation and washing, TMB substrate was added, and the OD450 was measured after 25 min. The neutralization index (IN) was calculated as: IN = 100 − (ODs/ODnc) × 100 (%), where ODs is the mean OD450 of the test sample and ODnc is the mean OD450 of the negative control. A positive result was defined as IN > 20%, consistent with the positive control (neutralizing antibody concentration of 12.5 PFU/mL).

##### Evaluation of Antiviral Efficacy

The antiviral efficacy of the samples was assessed following the guidelines of the Scientific Centre for Expert Evaluation of Medicinal Products, Ministry of Health of the Russian Federation.

The viral inhibition coefficient (CI, %) was calculated as: CI = [(Anc − As)/Anc] × 100 (%), where Anc is the virus concentration (PFU/mL) without test samples, and As is the virus concentration (PFU/mL) after adding test samples, both determined via plaque assay on Vero cell monolayers.

### 2.8. Construction of Recombinant Probiotic Vaccines and Recombinant Proteins

The construction of clones L3-S1 and L3-PSP has been previously reported [10,11]. The production of recombinant proteins S1 and PSP was carried out using a previously described method [10,11].

#### 2.8.1. S1 Protein

The S1 protein was produced in a recombinant *E. coli* strain under denaturing conditions. Bacteria were cultured in Terrific Broth (Himedia) with ampicillin (100 µg/mL) and kanamycin (25 µg/mL) until late logarithmic growth (OD600 = 0.7–0.9). IPTG was added to induce protein expression, followed by a 4.5 h incubation. Cells were harvested by centrifugation, and the pellet was frozen at −70 °C. After thawing, the pellet was resuspended in Buffer A (8 M urea, 0.1 M Na_2_HPO_4_, 0.1 M NaH_2_PO_4_, pH 8.0) and lysed for 1 h. The supernatant was purified using Ni Sepharose (Qiage, Shenzhen, China). The S1 protein eluted under denaturing conditions, displaying a single 24.5 ± 0.5 kDa band on Coomassie-Brilliant-Blue-stained 12% SDS-PAGE.

The protein was refolded by two-step dialysis: first against 3 M urea, 0.1 M Na_2_HPO_4_/NaOH (pH 9.2) for 2 h, then against 0.4 M NaCl, 0.02 M Na_2_HPO_4_/NaOH (pH 9.2) overnight at 6 °C. The refolded S1 protein was sterilized by filtration (0.45 µm, Millipore) and stored at 6 °C. MALDI TOF/TOF analysis (Bruker Daltonics) confirmed the sequence of the purified S1 protein as part of the SARS-CoV-2 S protein.

#### 2.8.2. PSP Protein

The synthetic chimeric gene encoding the PSP protein was inserted into pET25b+, creating the expression plasmid pPSP, which was transformed into *E. coli* BL21(DE3). Recombinant *E. coli* was grown in PYP5052 medium with 0.2% lactose. PSP proteins were extracted by lysing cells in 20 mM Tris-HCl (pH 7.5), 5 mM EDTA, and 1 mM PMSF, followed by sonication (7 rounds of 30 s pulses). Bacterial lysates were centrifuged, and the pellets were resuspended in 1 M urea, then 2 M urea, and centrifuged again. The pellets were dissolved in 20 mM Na-phosphate buffer with 0.5 M NaCl and 6 M guanidine-HCl (pH 8.0) and purified using Ni-NTA (Qiagen). Final dialysis was conducted against 20 mM Tris-HCl (pH 7.5), 5 mM EDTA, and 1 mM PMSF, followed by alkaline bidistilled water. The protein was either vacuum-dried or stored in aqueous solution at −80 °C.

### 2.9. Histological Analysis

#### 2.9.1. Hamsters

After collection, the lungs were fixed in 10% neutral formalin for 21 days at room temperature to ensure the inactivation of the virus. The samples were then washed three times with distilled water, with each wash lasting 1 h. Following this, the tissues were dehydrated using a series of ethanol solutions with increasing concentrations and embedded in paraffin (Richard-Allan Scientific Paraffin, Microm, Munich, Germany) using a Spin Tissue Processor STP 120 (Microm). Sections, each 5 microns thick, were cut from the paraffin blocks using a Rotary 3003 microtome (PFM Medical, Germany) and mounted onto HistoBond-M adhesive slides (Marienfeld, Germany).

The lung tissues were stained with hematoxylin and eosin, which resulted in blue-violet staining of the nuclei, moderately oxyphilic staining of the cytoplasm in smooth muscle cells, red-brown staining of erythrocytes, and pink coloration of connective tissue fibers. The specimens were examined under an Olympus CX41 microscope (Olympus, Tokyo, Japan), with images captured by a MIchrome 5 Pro digital camera (Tucsen, Fuzhou, China) under standardized light, contrast, and magnification settings.

#### 2.9.2. Mice

After extraction, the animals’ lungs were fixed by immersion in zinc ethanol-formaldehyde for 24 h at room temperature [13]. The lungs were then washed of the fixing mixture in 96% ethanol, after which the material was placed in histological cassettes, and further dehydration was carried out using a Microm STP 120 (Microm, Germany) automatic tissue processor. Paraffin embedding (Richard-Allan Scientific Paraffin, Microm, Germany) was performed using the standard technique. Sections, each 5 µm thick, from the right lung paraffin blocks were cut using a Rotary 3003 microtome (PFM Medical, Germany) and mounted on HistoBond-M adhesive slides (Marienfeld, Germany). The condition of the lung tissue was assessed using hematoxylin-eosin staining and an immunohistochemical reaction to the Iba-1 protein, for which recombinant rabbit monoclonal antibodies (clone JM36-62) were used at a dilution of 1:900 (ET-1705-78, Huabio, Hangzhou, China). The Reveal-HRP kit (Abcam, Cambridge, UK) was used for secondary antibodies. The reaction product was visualized using the chromogen 3′3-diaminobenzidine from the DAB+ kit (K3468, Agilent, Santa Clara, CA, USA). After the immunohistochemical reaction, the sections were counterstained with alum hematoxylin. Hematoxylin-eosin staining resulted in blue-violet nuclei, moderately oxyphilic smooth muscle cell cytoplasm, red-brown erythrocytes in the vessel lumen, and pink connective tissue fibers. The immunohistochemical reaction yielded a brown precipitate at the site of Iba-1 protein localization, with blue-violet nuclear staining. Microscopic examination and photography of the preparations were performed using an Olympus CX41 microscope (Olympus, Tokyo, Japan) and a MIchrome 5 Pro digital camera (Tucsen, Fuzhou, China). Micrographs were taken at consistent magnification with unchanged light and contrast settings.

### 2.10. Statistical Analyses

The Shapiro–Wilk test was employed to evaluate data normality, while statistical significance was assessed using the Student’s *t*-test. Results are expressed as the mean ± SEM. Differences among groups were analyzed using ANOVA with Tukey’s multiple comparison test or, for non-normally distributed data, the nonparametric Mann–Whitney U-test or Fisher’s exact test. The statistical analysis was conducted using GraphPad Prism 6 software (GraphPad Software, Inc., San Diego, CA, USA), with *p*-values < 0.05 regarded as significant. To assess the correlation between antigen-specific IgG and IgA, Pearson’s correlation was applied to matched saliva and serum/plasma samples collected from the same individual at the same time point.

### 2.11. Ethics Statement

All experiments were conducted in strict accordance with Directive 2010/63/EU of the European Parliament and of the Council of 22 September 2010 on the protection of animals used for scientific purposes Text with EEA relevance on the protection of animals used for scientific purposes, as well as the Federation of European Laboratory Animal Science Associations (FELASA) guidelines for health monitoring in mouse, rat, hamster, guinea pig, and rabbit colonies in breeding and experimental settings. The experiments involving hamsters were approved and performed in compliance with these regulations under the supervision of the local biomedical ethics committee at the Federal State Budgetary Institution, 48th Central Research Institute of the Ministry of Defense of the Russian Federation. This approval is documented in the meeting minutes dated 3 November 2021.

Approval for experiments involving mice was granted by the Local Ethics Committee at the Federal State Budgetary Institution ‘Institute of Experimental Medicine (IEM)’, as recorded in the meeting minutes from 28 January 2021 (meeting minutes 1/21).

## 3. Results

### 3.1. Study of the SARS-CoV-2 S1 Vaccine Antigen

Syrian hamsters were immunized according to the protocol described in Section 2 (Figure 1A). Half of the hamsters were immunized intramuscularly with the S1 protein with adjuvant (group S1), while the remaining hamsters served as untreated controls (group untreated control). The other half were orally vaccinated with the recombinant probiotic vaccine strain *E. faecium*, which expresses the S1 protein on the surface of the bacterium L3 (group L3-S1) [9]. Animals that received unmodified *E. faecium* L3 orally, according to the same protocol, served as controls (group L3). The recombinant S1 protein under investigation corresponds to a segment of the SARS-CoV-2 beta variant spike (S) protein, covering positions 496 to 646. This region includes part of the receptor-binding domain (RBD) and the SD2 subdomain (Figure 2) [11]. This region is crucial for the virus’s entry into host cells, making it a key target for interventions aimed at preventing and treating COVID-19 [16].

Twenty-one days after the second vaccination, the levels of S-specific antibodies were measured in the blood serum and buccal washes of hamsters immunized by two different methods. Two variants of the neutralization assay and ELISA were used. In the neutralization test, which assessed the suppression of negative colonies formed by SARS-CoV-2 in a one-day monolayer of Vero C1008 cells under an agar overlay, a low level of virus-neutralizing antibodies was detected following two oral administrations of the recombinant probiotic vaccine L3-S1. The reciprocal titer was calculated to be 2–4. In contrast, after two subcutaneous administrations of the recombinant protein S1, the reciprocal titer of virus-neutralizing antibodies in the serum of hamsters ranged from 2 to 21. Virus-specific neutralizing antibodies were not detected in the *E. faecium* L3 group or the untreated control group (Table 1).

When evaluating virus-neutralizing activity using an enzyme immunoassay, blood serum samples and nasopharyngeal swabs from immunized animals, collected on the 28th day post-immunization, were analyzed.

The assessment of virus-neutralizing activity, based on the inhibition of S protein binding to human ACE2 in ELISA, revealed that the majority of blood sera from animals immunized with the L3-S1 vaccine exhibited positive virus-neutralizing activity, unlike the control group (Figure 3; Appendix A). No positive virus-neutralizing activity was detected in the sera of animals parenterally immunized with the S1 protein. The virus-neutralizing activity of nasopharyngeal swabs from all studied groups was below the positive threshold (20%). However, in mice vaccinated with the probiotic, this activity correlated with the virus-neutralizing activity in the blood serum, with a correlation coefficient of 0.51, indicating a moderate positive relationship between the variables according to the Chaddock scale.

The level of specific IgG antibodies in individual blood sera was further assessed using an enzyme-linked immunosorbent assay (ELISA) with a commercial full-length S protein of SARS-CoV-2 (Figure 4A). Additionally, pooled samples of hamster sera from each group were tested in ELISA, using the recombinant S1 protein employed in vaccination adsorbed to the plate (Figure 4B). The indirect enzyme immunoassay revealed that specific antibodies accumulated in the blood of animals from both experimental groups following vaccination (Figure 4). The most significant differences in specific IgG levels were observed when using the original recombinant S1 protein in ELISA. In this case, the titer of serum antibodies in the blood of subcutaneously immunized hamsters was substantially higher compared to those after oral vaccination (Figure 4B). The circle represents individual data in each group.

The analysis demonstrated that both subcutaneous and oral vaccination with the S1 antigen elicit a specific systemic immune response in Syrian hamsters. A tendency was observed for higher levels of serum antigen-specific IgG following parenteral immunization and higher levels of serum virus-neutralizing antibodies after oral vaccination.

To evaluate the protective effectiveness of the induced immune response, vaccinated hamsters were orally infected with coronavirus at a dose of 4.3 log PFU in a volume of 200 μL. The concentration of the virus in the lungs was assessed on days 3 and 6 post-infection. The results of the infectious titer assessment of SARS-CoV-2 in lung tissue are presented in Figure 5.

Compared to the control, the maximum suppression of virus reproduction in the lungs of parenterally vaccinated animals was observed on day 3 post-infection, while orally vaccinated hamsters exhibited the greatest inhibitory effect compared to the control on day 6 post-infection. In both cases, the inhibition coefficients were similar, at 83.7% and 82.1%, respectively.

Histological analysis Figure 6a(A,B) revealed that hematoxylin and eosin staining of lung tissues from intact animals showed no evidence of pathological processes, including pneumonia foci, peribronchial infiltration, vasculitis, or dystrophic changes in the bronchial epithelium. In the lungs of animals in the untreated control group, a reduction in lung airiness was observed on the third day post-infection, which persisted throughout the observation period. While diffuse alveolar changes were present, they were not associated with the formation of pneumonia foci.

In the S1 group, a similar reduction in lung airiness was noted in all animals post-infection, along with diffuse alveolar changes and small pneumonia foci containing fibrinous-hemorrhagic exudate. However, no accumulation of neutrophilic granulocytes was detected in the pneumonia foci (Figure 6a(A)). In the experimental L3 group, reduced lung airiness persisted post-infection, accompanied by diffuse alveolar changes with fibrinous-hemorrhagic exudate and the emergence of pneumonia foci by the sixth day. Notably, no neutrophilic granulocyte accumulation was found in the pneumonia foci. Signs of productive vasculitis were evident, characterized by mononuclear and macrophage infiltration of the vascular adventitia.

In the L3-S1 group, the observed lung changes were less severe throughout the study period; however, a decrease in lung airiness was still present. Diffuse alveolar changes, including areas of atelectasis with fibrinous-hemorrhagic exudate, were noted.

Thus, the histological analysis performed when comparing groups S1 and L3-S1 on the third and sixth days after infection demonstrated a more pronounced inflammatory reaction in animals that received parenteral vaccination (group S1), indicating a less pronounced anti-inflammatory effect of the parenteral vaccine compared to the mucosal vaccine.

### 3.2. Study of the PSP Vaccine Antigen

Balb/c mice were immunized using the traditional route of subcutaneous administration of the PSP protein in the presence of aluminum hydroxide. As an alternative, the route of oral administration of the probiotic strain *E. faecium* L3 carrying the same protein was used. The immunization schedule is provided in the Section 2 (Figure 1B). The control groups consisted of animals that received subcutaneous adjuvant and the original variant of *E. faecium* L3, respectively.

Local and systemic immune responses to vaccination were assessed by ELISA. Enzyme-linked immunosorbent assay of secretory IgA levels in nasal and oral swabs showed that, 14 days after the start of immunization, a significant increase in PSP-specific antibodies was observed in the group of animals receiving the probiotic vaccine (Figure 7A).

The level of specific immunoglobulin G in the blood serum showed a significant increase only in the group of mice that received subcutaneous immunization 14 days after the start of immunization (Figure 7B). Four weeks after the second subcutaneous immunization, there was a further increase in antigen-specific serum IgG levels. By this time, the levels of PSP-specific IgG antibodies in the sera of orally immunized mice also increased significantly, though at notably lower titers compared to subcutaneous immunization (Figure 7C).

The results of the ELISA analysis of serum PSP-specific IgA antibodies following the first immunization are noteworthy. No specific IgA antibodies were detected in any of the groups. However, significant differences were observed in the antigen-specific binding of IgA in the sera of orally and parenterally immunized mice. After a single cycle of oral vaccination, sera from both control and experimental groups demonstrated significantly higher IgA binding to the PSP antigen in ELISA assays (Figure 8A).

A comparative analysis of sera from the control groups (L3 and ad) and untreated animals indicated that both subcutaneous adjuvant administration and oral administration of the unmodified *E. faecium* L3 probiotic strain resulted in the accumulation of class A antibodies in the blood, capable of binding to the PSP antigen. Notably, this accumulation was more pronounced following oral administration of the probiotic bacteria (Figure 8B).

We hypothesize that vaccination not only promotes the accumulation of specific IgA but also induces non-specific IgA with a broad spectrum of activity. This non-specific IgA may bind to various pathogens non-selectively, potentially facilitating their elimination in a manner similar to specific antibodies. To test this hypothesis, we evaluated the protective potential of sera obtained after the first vaccination cycle in a passive protection assay against *S. pneumoniae* infection.

BALB/c mice were intranasally infected with a suspension of *S. pneumoniae* at a dose of 4 × 10^5^ CFU. Before infection, the bacteria were incubated with sera pooled from control and vaccinated mice from experimental groups subjected to oral (L3, L3-PSP) and parenteral (ad, ad + PSP) immunization (Figure 9) and untreated intact mice.

Twenty-four hours after infection, the *S. pneumoniae* load in the lungs was measured. Results indicate that sera from intact mice and mice receiving only the adjuvant did not influence pneumococcal growth in the lungs. In contrast, sera from mice immunized subcutaneously with PSP (PSP group) and those from the probiotic vaccine (L3-PSP group) provided significant protection against pneumococcal infection compared to sera from untreated mice. A comparison between the untreated and L3 groups suggests a trend toward increased protective potential of sera from animals receiving the probiotic strain *E. faecium* L3 orally.

A comparative assessment of the protective efficacy of the two vaccination methods was conducted through an experiment on active protection of vaccinated mice against intranasal *S. pneumoniae* infection (Figure 10).

Fifty-two days after the initial immunization and one month following the second administration of the vaccines, mice were intranasally infected with a suspension of *S. pneumoniae* at a dose of 10^5^ CFU. The bacterial load in the lungs was measured 24 and 48 h after infection. Notable differences were observed 24 h post-infection. In both control groups, five out of six animals exhibited a high bacterial load in the lungs. In contrast, in the immunized groups (both oral and parenteral), only two out of six mice had detectable pneumococci in the lungs. Fisher’s exact test revealed a *p*-value of 0.242 for both vaccination methods, indicating no significant difference between the immune and non-immune groups. By 48 h post-infection, most mice in both the immune and control groups had cleared the infection from their lungs. Some mice were retained for further assessment of the effect of vaccination on mortality rates from pneumococcal infection (Table 2).

Mortality rates were recorded 10 days after intranasal infection. All animals in the parenterally and orally vaccinated groups survived, whereas mortality in the control groups was 40% and 20%, respectively.

Histological analysis of lung tissue from intact mice revealed no signs of pneumonia, peribronchial infiltration, vasculitis, or other pathological processes when stained with hematoxylin and eosin. In the lungs of animals from the adjuvant and adjuvant-PSP groups, lung airiness was maintained throughout the observation period (1, 2, and 5 days post-infection), with no evidence of pneumonia foci. The most notable changes were observed in the peribronchial connective tissue, characterized by perivascular edema and infiltration of mononuclear cells and macrophages (Figure 11). No significant alterations were detected in the bronchial epithelium or vascular endothelium.

On the second day post-infection, diffuse alveolar changes were noted, accompanied by a proliferative reaction in the interalveolar septa and subpleural zone. By the fifth day, bronchitis was observed in small and medium bronchi, characterized by the presence of purulent contents within the bronchial lumen, without a corresponding reaction from the bronchial epithelium.

In the lungs of animals in the L3 and L3-PSP groups, lung airiness was preserved throughout the observation period (1, 2, and 5 days post-infection), with no development of pneumonia foci. Changes in the peribronchial connective tissue were minimal, marked by mild perivascular edema and infiltration of mononuclear cells and macrophages. Additionally, no significant changes were observed in the bronchial epithelium or vascular endothelium. Multiple focal proliferative reactions were recorded in the subpleural region and adjacent interalveolar septa on the 2nd and 5th days post-infection.

In conclusion, the histological data confirmed the effectiveness of both vaccination methods, as both resulted in a reduction in the severity of morphological signs of infection-induced inflammation.

The results of this study suggest that intramuscular immunization of Syrian hamsters with the S1 protein, as well as mice with the PSP protein via subcutaneous administration with an adjuvant or orally as part of a live probiotic vaccine, induces a specific systemic immune response. Despite notable differences in the humoral components, this response provides comparable protection against coronavirus and pneumococcal infections, respectively.

## 4. Discussion

This study compared the effectiveness of two vaccination routes, the conventional parenteral route and the less-explored mucosal route, in terms of stimulating the humoral immune response and providing protection against viral and bacterial infections in laboratory animals. The models used were coronavirus infection in hamsters and pneumococcal infection in mice. The vaccine antigens used were a recombinant fragment of the coronavirus spike protein S1 and a recombinant chimeric protein (PSP), which consists of amino acid sequences corresponding to immunogenic regions of three protein virulence factors of *S. pneumoniae*, which were previously obtained and studied [10,11]. The bacterial vector used for delivering the vaccine antigen to the gastrointestinal mucosa was the well-characterized probiotic strain *E. faecium* L3. Using a method developed earlier, we inserted DNA sequences encoding the recombinant proteins under study into the *E. faecium* L3 genome. These sequences were incorporated into the gene sequence of the enterococcal major pili protein and expressed on the surface of the probiotic bacterium [11,13]. This bacterial vector addressed the critical issue of protecting the vaccine antigen from degradation in the gastrointestinal tract and ensured its prolonged presence on the mucosa through bacterial proliferation. We have previously demonstrated that *E. faecium* L3 strains modified with vaccine antigens can persist in the gastrointestinal tract for up to 10 days following oral administration [13].

Cloning of the S1 and PSP DNA fragments in *E. coli* enabled the production of the corresponding recombinant proteins, which were then used as parenterally administered structural analogues of the antigens present in enterococci.

Coronavirus infection was studied using a model of Syrian hamsters. Despite significant differences in the humoral immune responses of Syrian hamsters following immunization via the two different routes, a comparable level of protection against coronavirus infection was achieved with both parenteral and oral administration of the S1 antigen vaccine.

It was anticipated that intramuscular immunization would elicit a stronger humoral immune response, as parenteral administration facilitates immediate contact with lymphocytes throughout the body due to the vaccine material entering the bloodstream. In contrast, vaccination via the gastrointestinal mucosa typically induces a primarily local reaction. Over time, with the prolonged presence of the antigen on the mucosa, activation of T- and B-cells occurs, leading to a systemic adaptive immune response through the migration of antigen-presenting cells to the lymph nodes [17,18].

Indeed, our analysis of virus-neutralizing antibody levels using the plaque reduction neutralization test on Vero cells, along with serum IgG antibodies specific to the commercial S protein of the coronavirus and the recombinant S1 protein in indirect ELISA, indicated a stronger humoral immune response in hamsters following parenteral vaccination (Table 1 and Figure 4). However, when evaluating virus-neutralizing activity in Syrian golden hamsters by inhibiting protein S binding to human ACE2 in ELISA, only sera from orally immunized hamsters exhibited such activity (Figure 3, Appendix A). This discrepancy may be attributed to the assay’s detection of not only specific immunoglobulins but also broad-spectrum IgA, which, according to various studies, increases in the blood following the intake of probiotic bacteria [19,20,21,22].

This assumption is supported by the data from the present study (Figure 3, Appendix A), which show that the inhibition of protein S binding by sera from control hamsters receiving *E. faecium* L3 is equivalent to that observed in mice parenterally vaccinated with protein S1.

In the pneumococcal bacterial infection model using Balb/c mice vaccinated with pneumococcal protein PSP, a similar conclusion was reached regarding the comparable effectiveness of immune protection following both parenteral and oral vaccination with the same antigen. While parenteral vaccination resulted in a more rapid and intense increase in PSP-specific IgG antibody levels, this did not translate into superior protection against intranasal pneumococcal infection. Oral vaccination, despite inducing a significantly lower level of specific serum IgG response at the time of infection, provided comparable protection against pneumococcal infection, as assessed by bacterial load in the lungs (Figure 10) and mortality rates (Table 2).

Potential reasons for the observed equivalence in protection, despite significant differences in systemic humoral immune response levels, may involve other immune factors not assessed in this study. For viral infections, systemic cellular immunity and gastrointestinal mucosal immunity could be relevant [23,24,25]. For pneumococcal infection, the complementary role of innate immunity mechanisms, particularly the protective role of natural antibodies, should be considered.

This is illustrated by our observations of the serum properties obtained two weeks after the initiation of oral vaccination in mice (Figure 7B, Figure 8 and Figure 9). According to the indirect ELISA data, antigen-specific IgG and IgA were absent in the sera (Figure 7B and Figure 8A), with no differences observed between the control and immune groups. However, comparing IgA antibody titers binding the PSP antigen at the bottom of the plate revealed significant differences between mice immunized parenterally and those immunized orally. These differences were observed in both the vaccinated and corresponding control groups (Figure 8A), suggesting a non-specific interaction. Comparison of the “L3” and “ad” control groups with sera from normal, untreated mice indicates that the accumulation of circulating non-specific IgA antibodies is most pronounced following oral administration of *E. faecium* L3 (Figure 8B). A single administration of aluminum hydroxide as an adjuvant also resulted in a significant, though lesser, increase in the level of serum IgA capable of binding the PSP antigen at the bottom of the plate.

We hypothesized that such antibodies might also interact with pneumococci and contribute to reduced infectivity, potentially through opsonization. To test this hypothesis, we infected normal mice intranasally with *S. pneumoniae* after pre-incubating the inoculum with sera from the experimental groups and, for comparison, with serum from intact untreated animals. Indeed, passive protection led to significant differences in bacterial accumulation in the lungs 24 h after infection. Sera from untreated mice and control mice that received the adjuvant subcutaneously had minimal impact on pneumococcal proliferation. In contrast, serum from mice immunized with PSP subcutaneously completely suppressed bacterial proliferation, correlating with the presence of PSP-specific IgG (Figure 7B). Notably, the serum from mice immunized orally with L3-PSP also inhibited bacterial proliferation, despite the absence of detectable PSP-specific antibodies according to ELISA. Interestingly, serum from mice receiving unmodified *E. faecium* L3 exhibited antibacterial activity, although these results did not reach a statistically significant difference from the untreated control.

This suggests that using probiotic bacteria as vectors for antigen delivery to mucosal surfaces not only addresses the critical issue of antigen stability necessary for inducing an immune response [26,27,28] but may also enhance innate immunological protective factors, such as natural antibodies, which can act early in the infectious process [29,30,31]. This may explain the positive results observed with probiotic bacteria in humans for enhancing overall resistance during seasonal disease outbreaks [22,32,33], as well as the increasing focus by vaccinologists on the role of polyfunctional antibodies and their impact on vaccination outcomes [34].

Our histological analysis of the lungs from vaccinated and control animals during infection indicates that the intensity of the inflammatory response, at least in the early stages of the infection process, is less pronounced with oral vaccination (Figure 6 and Figure 11). The severity of the inflammatory process is influenced by the interplay between pathogen characteristics and host immune responses. The specific immune response to the antigen plays a crucial role in modulating the inflammatory reaction, which aids in controlling the infection while also contributing to tissue damage [35]. In the experiments described, it appears that the oral vaccination route achieves the optimal balance between these two processes.

Our observations were recorded one month after the completion of vaccination. Questions regarding whether this pattern will persist over longer periods post-immunization and the duration of immunological memory for each vaccination method remain open and will be addressed in future studies.

The results suggest that parenteral vaccination methods could potentially be supplemented or, in some cases, replaced by probiotic strain vaccination, which offers cost advantages for manufacturers and may be more appealing to consumers [36,37]. This approach could be particularly beneficial for immunologically vulnerable populations, such as children, the elderly, and patients with comorbid conditions [34]. Consequently, an urgent question arises: should specific IgG accumulation be considered the sole gold standard for evaluating vaccine effectiveness, or should alternative criteria be employed for assessing mucosal vaccination quality [1]? The increasing number of publications on recombinant probiotic vaccines reflects a sustained interest in using probiotic bacteria as live vectors for delivering various vaccine antigens to mucosal surfaces [38,39]. However, the challenge of interpreting standard immunological analyses for evaluating such vaccines is noted [36]. Resolving this issue is crucial to advancing the development of non-parenteral, particularly recombinant probiotic, vaccines beyond preclinical trials.

## 5. Conclusions

A comparative study was conducted on experimental animals to evaluate the protective efficacy of parenteral versus mucosal vaccination routes. The same antigen was delivered either via parenteral injection or through the mucosal membrane using oral administration of a recombinant probiotic vaccine. The results demonstrated equivalent efficacy of both vaccination approaches using viral and bacterial vaccine proteins, and the histological findings are consistent with this conclusion. These findings suggest that developing mucosal probiotic vaccines could be a viable, cost-effective, and safe approach for preventing viral and bacterial infections.

## Figures and Tables

**Figure 1 vaccines-12-01195-f001:**
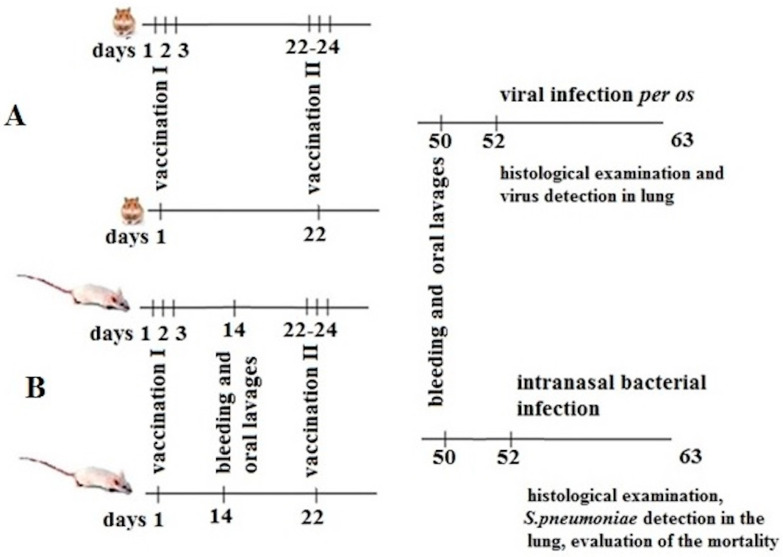
Experimental design for evaluating vaccination efficacy. (**A**) Study of the vaccine strain L3-S1 in hamsters. (**B**) Study of the vaccine strain L3-PSP in mouse.

**Figure 2 vaccines-12-01195-f002:**
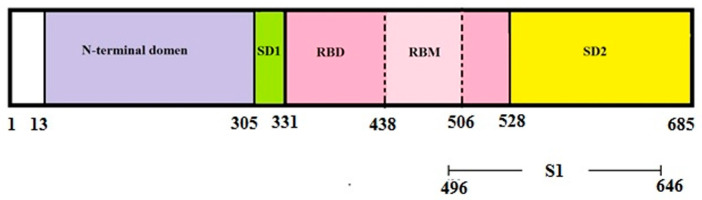
Recombinant protein S1 within the N-terminal S1 ectodomain of the SARS-CoV-2 spike (S) protein: RBD (receptor-binding domain; residues 319–527); RBM (receptor-binding motif; residues 438–506); SD1 and SD2 (S1 and S2 subdomains).

**Figure 3 vaccines-12-01195-f003:**
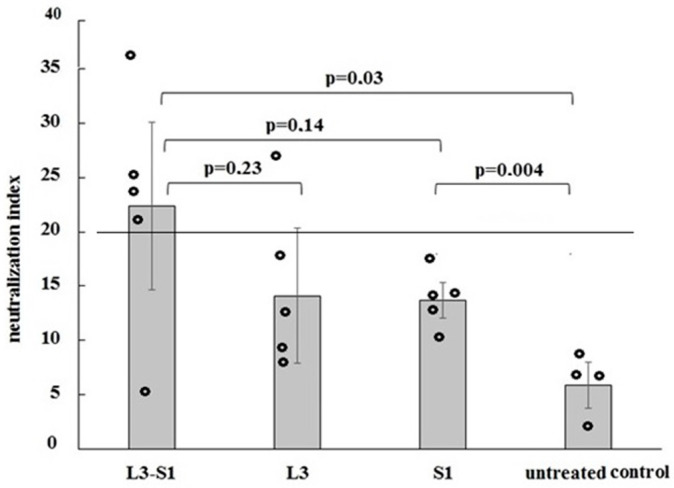
Evaluation of virus-neutralizing activity in Syrian golden hamsters by inhibition of S protein binding to human ACE2 in ELISA. A neutralization index greater than 20% was considered positive. Statistical analysis of group means was performed using one-way ANOVA followed by pairwise comparisons with the *t*-test.

**Figure 4 vaccines-12-01195-f004:**
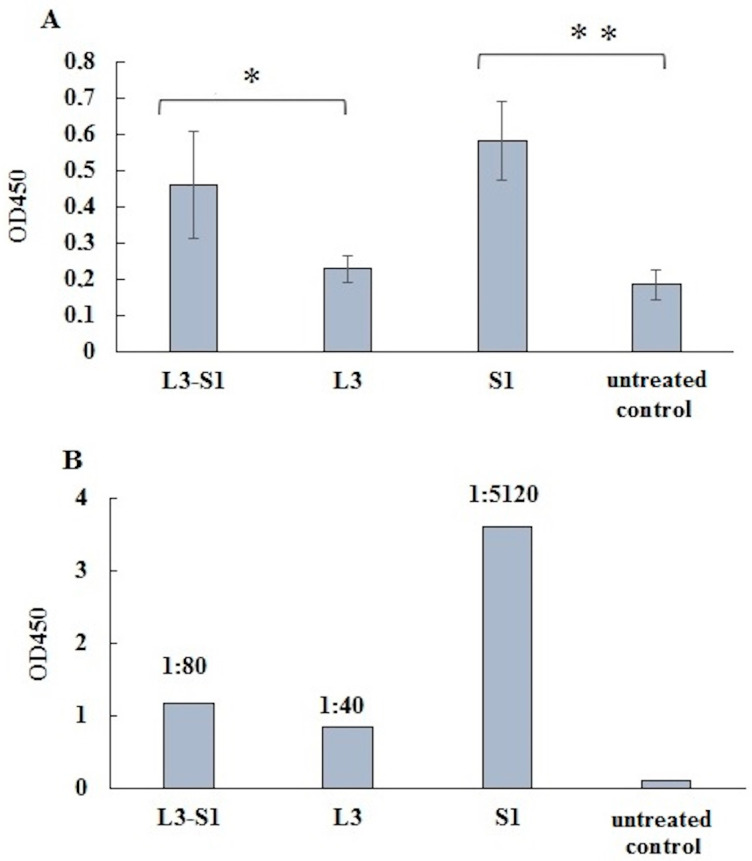
(**A**) Analysis of specific IgG levels in blood sera by indirect ELISA. The commercial full-length S protein of SARS-CoV-2 was adsorbed onto the plate. Individual sera were diluted 1:100, and OD450 was measured. Results are presented as the mean ± SEM (*n* = 5/group). A one-way ANOVA was used to compare the means of the groups. Statistically significant differences between groups were identified using the *t*-test. * *p* < 0.05; ** *p* < 0.01. (**B**) The original recombinant S1 protein was adsorbed onto the plate. Hamster sera from each group were pooled in equal proportions, and the antibody titer was determined for each serum pool. The figure shows OD450 values at a 1:20 dilution and serum titers.

**Figure 5 vaccines-12-01195-f005:**
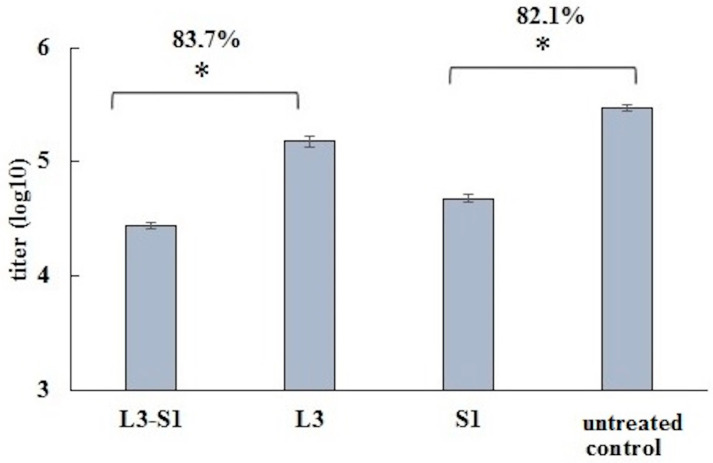
Analysis of infectious virus accumulation in the lungs of Syrian hamsters after infection. The figure displays the percentage of virus reproduction inhibition on days 3 (S1 and untreated control) and 6 (L3-S1 and L3) post-infection. * A statistically significant (*p* < 0.05) difference between groups was determined using the Mann–Whitney U test.

**Figure 6 vaccines-12-01195-f006:**
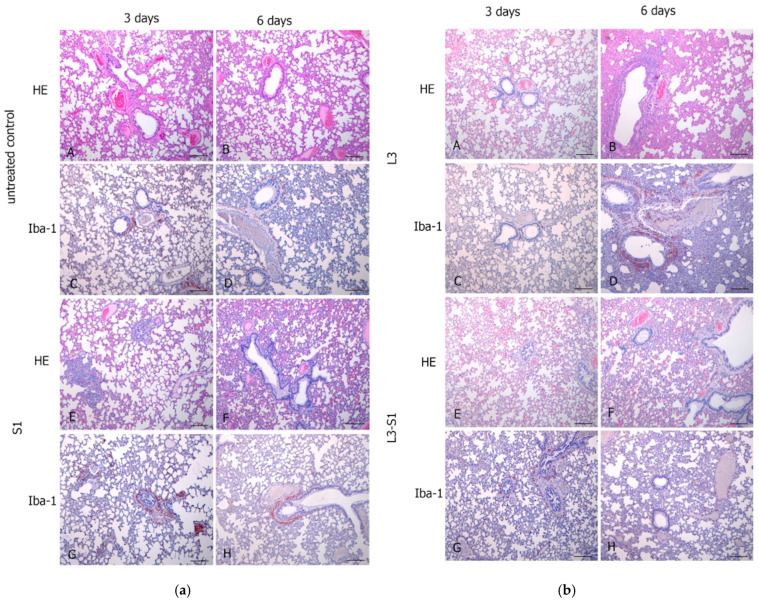
(**a**). Histological analysis of Syrian hamster lungs on days 3 and 6 post-SARS-CoV-2 infection. Lung tissue sections from unvaccinated animals at 3 days (**A**,**C**) and 6 days (**B**,**D**) post-infection, and from animals intramuscularly immunized with the S1 protein and adjuvant at 3 days (**E**,**G**) and 6 days (**F**,**H**) post-infection are shown. The intensity of the reaction highlights the accumulation of Iba-1 protein in macrophages, essential for phagocytic function (**C**,**D**,**G**,**H**). Macrophage infiltration is observed in the peribronchial and perivascular spaces, as well as in the lung parenchyma. Hematoxylin-eosin (HE) staining (**A**,**B**,**E**,**F**). Scale bar: 100 µm. (**b**). Histological analysis of hamster lungs on days 3 and 6 post-SARS-CoV-2 infection. Lung tissue sections from animals administered unmodified *E. faecium* L3 orally at 3 days (**A**,**C**) and 6 days (**B**,**D**) post-infection, and from animals receiving oral administration of the live recombinant probiotic vaccine L3-S1 at 3 days (**E**,**G**) and 6 days (**F**,**H**) post-infection are presented. The intensity of the reaction highlights the accumulation of Iba-1 protein in macrophages, crucial for phagocytic function. Macrophage infiltration is evident in the peribronchial and perivascular spaces (**C**,**D**,**G**,**H**). Hematoxylin-eosin (HE) staining (**A**,**B**,**E**,**F**). Scale bar: 100 µm.

**Figure 7 vaccines-12-01195-f007:**
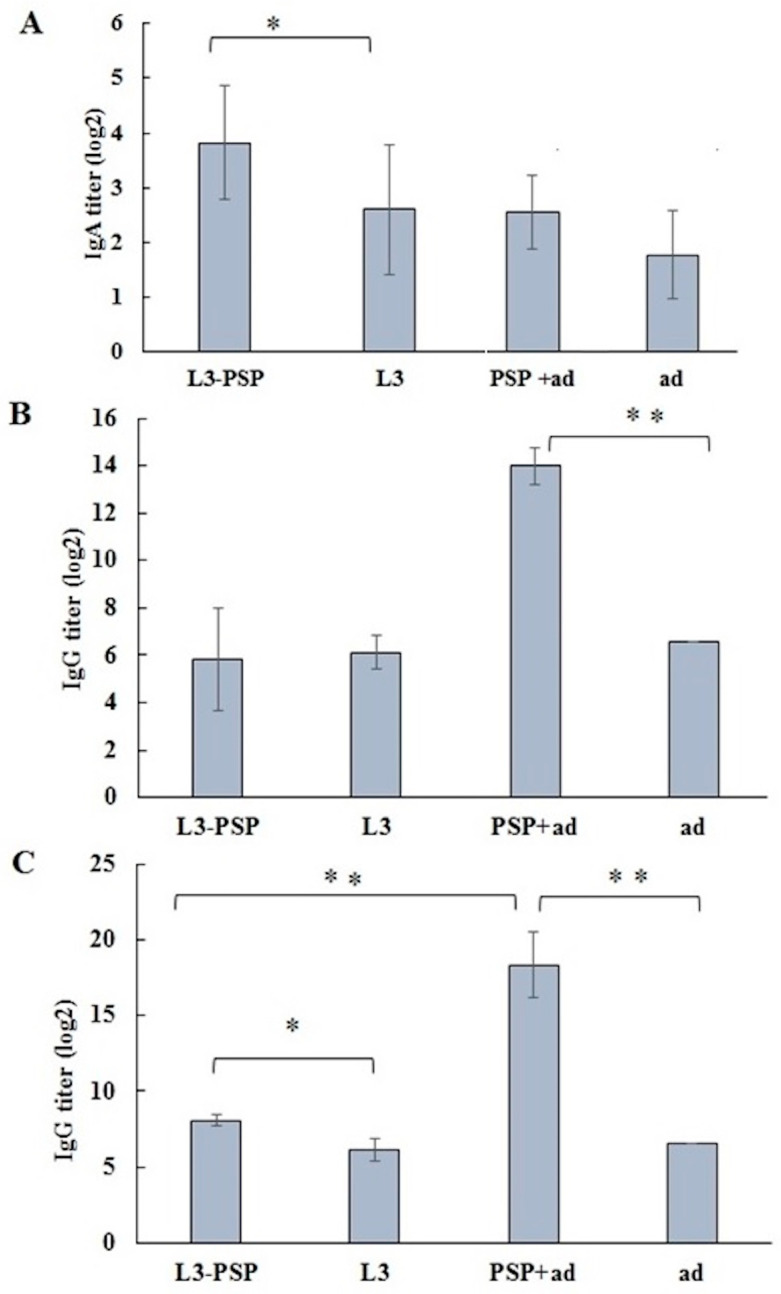
Enzyme-linked immunosorbent assay (ELISA) for specific antibodies in mice. (**A**) Secretory IgA in nasal and oral swabs 14 days after the start of immunization. (**B**) Specific IgG in mouse sera 14 days after the start of vaccination. (**C**) Specific IgG in mouse sera 50 days after the start of vaccination. Results are presented as the mean ± SEM (*n* = 5/group). Statistical analysis of the group means was performed using one-way ANOVA followed by pairwise comparisons using the *t*-test (* *p* < 0.05; ** *p* < 0.01).

**Figure 8 vaccines-12-01195-f008:**
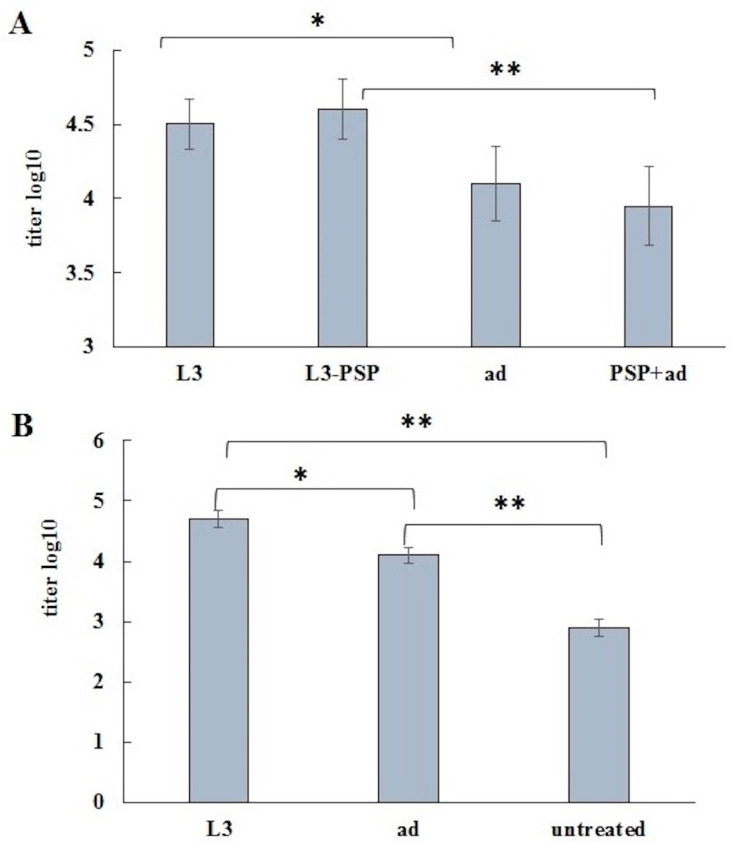
Enzyme-linked immunosorbent assay (ELISA) of IgA in mouse sera. (**A**) Analysis of PSP-specific IgA binding in the sera of control and vaccinated mice 14 days after the start of immunization. (**B**) Comparative analysis of PSP-specific IgA binding in control sera 14 days after the start of the experiment versus untreated mice. Statistical analysis of the group means was performed using one-way ANOVA followed by pairwise comparisons using the *t*-test (* *p* < 0.05; ** *p* < 0.01).

**Figure 9 vaccines-12-01195-f009:**
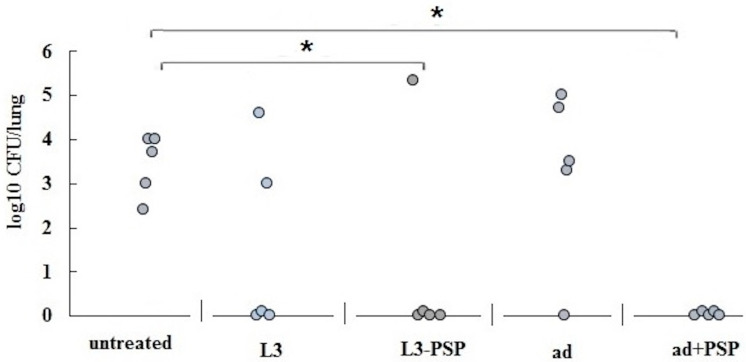
Passive protection of mice against intranasal S. pneumoniae infection. Normal Balb/c mice were intranasally infected with *S. pneumoniae* at a dose of 4 × 10^5^ CFU following a 30 min incubation of the bacteria with the studied sera at 37 °C. Twenty-four hours post-infection, the bacterial load of *S. pneumoniae* in the lungs was assessed. The significance of differences between groups was evaluated using Fisher’s exact test (* *p* < 0.05 indicates significant differences).

**Figure 10 vaccines-12-01195-f010:**
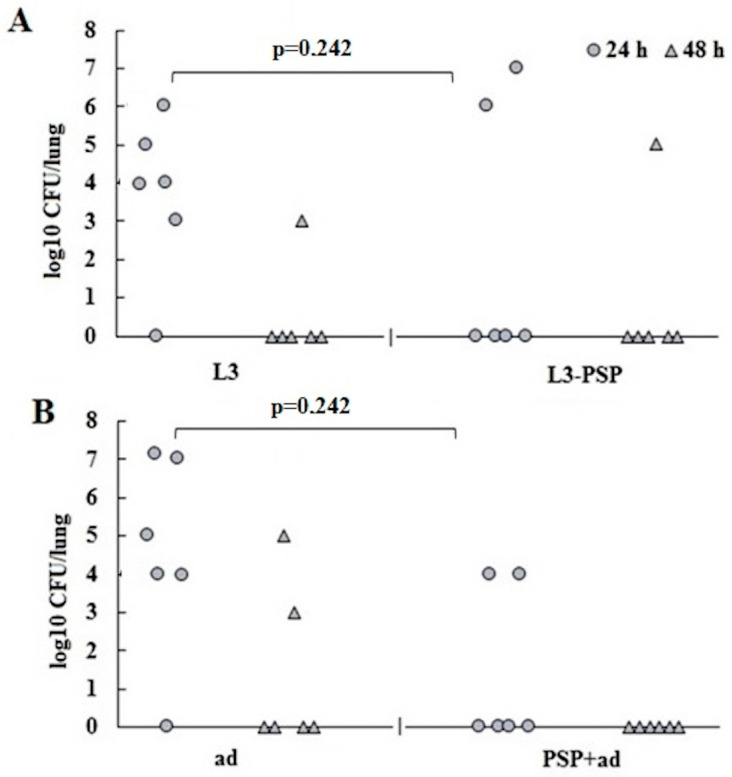
Determination of *S. pneumoniae* load in the lungs 24 and 48 h post-infection. (**A**) oral, (**B**) parenteral vaccination. Vaccinated and control mice were intranasally inoculated with 25 µL of *S. pneumoniae* at a dose of 10^5^ CFU one month after the second immunization. Lung samples from five mice per group were collected at 24 and 48 h post-inoculation to assess bacterial burden. Differences between groups were evaluated using Fisher’s exact test.

**Figure 11 vaccines-12-01195-f011:**
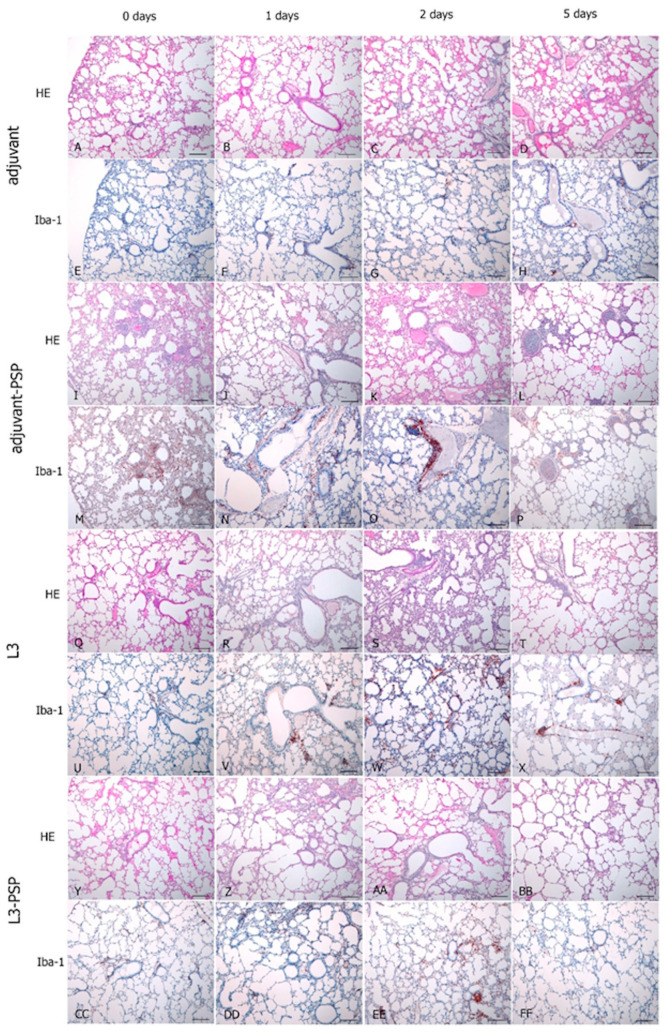
Histological analysis of mouse lungs following *Streptococcus pneumoniae* infection. Lung sections from animals in the adjuvant group are shown at day 0 (**A**,**E**), day 1 (**B**,**F**), day 2 (**C**,**G**), and day 5 (**D**,**H**) post-infection. Lung sections from the adjuvant-PSP group are displayed at day 0 (**I**,**M**), day 1 (**J**,**N**), day 2 (**K**,**O**), and day 5 (**L**,**P**) post-infection. The intensity of the Iba-1 protein accumulation in macrophages, which is essential for phagocytic function, is highlighted (**E**–**H**,**M**–**P**). Lung sections from animals in the L3 group are presented at day 0 (**Q**,**U**), day 1 (**R**,**V**), day 2 (**S**,**W**), and day 5 (**T**,**X**) post-infection. Lung sections from the L3-PSP group are shown at day 0 (**Y**,**CC**), day 1 (**Z**,**DD**), day 2 (**AA**,**EE**), and day 5 (**BB**,**FF**) post-infection. The Iba-1 protein accumulation in macrophages is indicated (**U**–**X**,**CC**–**FF**). Sections were stained with hematoxylin and eosin (HE) and subjected to immunohistochemical staining for Iba-1. Scale bar: 100 µm.

**Table 1 vaccines-12-01195-t001:** Level of virus-neutralizing antibodies in Syrian golden hamsters as determined by the plaque reduction neutralization test on Vero cells.

Experimental Groups	Number of Animals	Reciprocal Titer
L3-S1	5	(2–4)
S1	5	(2–21)
L3	5	˂2
Untreated control	5	˂2

**Table 2 vaccines-12-01195-t002:** Mortality of mice 10 days after intranasal pneumococcal infection.

Groups	Ad (*n* = 5)	PSP + ad (*n* = 5)	L3 (*n* = 5)	L3-PSP (*n* = 5)
Mortality (%)	40	0	20	0

## Data Availability

The data presented in this study are available on request from the corresponding author.

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
