# Peer review of "Comparative Efficacy of Parenteral and Mucosal Recombinant Probiotic Vaccines Against SARS-CoV-2 and S. pneumoniae Infections in Animal Models"

_vaccines, 2024, doi:10.3390/vaccines12101195_

Round 1
Reviewer 1 Report
Comments and Suggestions for Authors
This study compares the efficacy of parenteral and mucosal recombinant probiotic vaccines against SARS-CoV-2 and S. pneumoniae infections in animal models, presents intriguing findings, and explores an important area of vaccine development with well-designed animal studies, novel use of probiotic bacteria as vaccine antigen delivery vehicles, and comparative analysis of immune responses. However, several major and minor issues need to be addressed before the manuscript can be considered for publication, particularly given its potential to demonstrate comparable protection from probiotic-based mucosal vaccines, despite lower antibody levels.
Major Comments:
1. Image Quality: The resolution of the figures is inadequate, making them difficult to interpret. Please provide high-resolution images of all figures to ensure clarity.
2. Recombinant Protein Preparation: The Methods section lacks a detailed description of recombinant protein preparation. Please provide a comprehensive account of how the recombinant proteins were produced and purified.
3. Statistical Analysis: For Figures 3 and 4, t-tests were inappropriate. Please reanalyze the data using one-way ANOVA. Similarly, for Figures 7 and 8, replace the Mann-Whitney test with ANOVA for a more robust statistical analysis.
4. Data Presentation: In Figure 4, presenting the data as OD values is not ideal. Please provide endpoint titers or sigmoid curves to better represent the antibody levels.
5. The use of "PSP" is ambiguous. When referring to pneumococcal surface protein A (PspA), please use the correct terminology throughout the manuscript.
6. This study would benefit from a more comprehensive analysis of immune responses, including T cell responses and mucosal antibodies. This provided a more complete picture of the protective mechanisms involved.
7. The current study only examined short-term protection. This study aimed to assess the durability of immune protection over a longer period of time.
Minor Comments:
1. In the Materials and Methods section, the subheaders for "Immunization and Evaluation of Immune Protection" and "Evaluation of Passive Immune Protection" are missing. Please add these details for an improved organization.
2. The manuscript contains numerous typographical errors, missing spaces, and incorrect subscripts/superscripts. Please carefully proofread the manuscript.
3. Standardize terminology throughout the study. For instance, choose either "Fetal calf serum" or "FBS" and use it consistently.
4. Ensure that scientific names, such as Streptococcus pneumoniae, are spelled out in full at first mention and italicized appropriately.
5. Provide a rationale for using the beta variant S1 fragment and consider discussing how results might differ with the current strains.
Suggestions for Improvement.
1. Consider dose optimization studies for both parenteral and oral routes to strengthen the comparative analysis.
2. The impact of different oral dosing schedules on immune responses was explored.
3. Provide more mechanistic insights into how oral vaccination provides protection despite lower antibody levels.
4. Consider examining combinations of parenteral and oral vaccinations as a potential strategy to enhance overall protection.
Author Response
We sincerely thank the reviewer for their valuable comments and constructive feedback, and we appreciate the thoughtful and supportive review of our article.
Reviewer 1
Major Comments:
- Image Quality: The resolution of the figures is inadequate, making them difficult to interpret. Please provide high-resolution images of all figures to ensure clarity.
Reply.
It was done.
All Figures were provided at a sufficiently high resolution (more than 1000 pixels width/height and a resolution of 300 dpi). Common formats is JPEG.
- Recombinant Protein Preparation: The Methods section lacks a detailed description of recombinant protein preparation. Please provide a comprehensive account of how the recombinant proteins were produced and purified.
Reply.
Section 2.9 of the Methods section has been supplemented with a detailed description of the production and purification of recombinant proteins.
- Statistical Analysis: For Figures 3 and 4, t-tests were inappropriate. Please reanalyze the data using one-way ANOVA. Similarly, for Figures 7 and 8, replace the Mann-Whitney test with ANOVA for a more robust statistical analysis.
Reply.
Figure 3. A one-way ANOVA was used to compare the means of groups under investigation. The ANOVA showed significant differences between treatments, F= 3.57, p = 0.04. Tukey’s post-hoc test showed that group L3-S1 differed significantly from untreated control (p< 0.05) and group S1 differed significantly from untreated control (< 0.01).
The caption for Figure 3 has been changed.
Figure 4A. A one-way ANOVA was used to compare the means of the groups under investigation. The ANOVA showed significant differences between treatments (F = 11.93, p = 0.0002). Tukey’s post-hoc test revealed that group L3-S1 differed significantly from L3 (p < 0.05), and group S1 differed significantly from the untreated control (p < 0.01). The caption for part A of Figure 4 was modified.
Figure 7. A one-way ANOVA was used to compare the means of the groups under investigation. The ANOVA showed significant differences between treatments: Fig. 7A (F = 3.398, p = 0.037); Fig. 7B (F = 32.78, p = 0.00000007); Fig. 7C (F = 63.39, p = 0.00000000424). Tukey’s post-hoc test was used for pairwise comparisons, and the results are reflected in the figure. The caption for Figure 7 was modified.
Figure 8. A one-way ANOVA was used to compare the means of the groups under investigation. The ANOVA showed significant differences between treatments: Fig. 8A (F = 6.65, p = 0.003); Fig. 8B (F = 15.5, p = 0.0002). Tukey’s post-hoc test was used for pairwise comparisons, and the results are reflected in the figure. The caption for Figure 8 was modified.
- Data Presentation: In Figure 4, presenting the data as OD values is not ideal. Please provide endpoint titers or sigmoid curves to better represent the antibody levels.
Reply.
We agree that presenting data as OD values is not ideal. This choice was made due to the unique nature of the humoral immune response to this type of vaccine preparation, which consists of a probiotic vector and an antigen of interest. As a result of the oral administration of the probiotic vector, both control and vaccinated animals accumulate broad-spectrum IgA antibodies, which are capable of non-specifically binding to antigens at the bottom of the ELISA plate (discussed in this article, see Fig. 8). In addition, mice vaccinated with recombinant probiotic vaccines have broad-spectrum IgG antibodies, as discussed in our previous paper, where titration curves are provided. (doi: 10.3390/ijms25010215. Thus, control sera contain polyfunctional antibodies, while immune sera contain both antigen-specific and polyfunctional antibodies. Consequently, in some cases, the sigmoid curves for titration of immune sera in ELISA closely resemble those of the controls but are higher due to the presence of antigen-specific antibodies. Therefore, from our perspective, comparing OD values in the hamster experiment allows us to assess the presence of an antigen-specific immune response. This method of comparison is frequently used by researchers studying the humoral immune response to mucosal vaccination (e.g., Wang et al., 2023, doi: 10.1128/spectrum.00102-23; Jia et al., 2022, doi: 10.3389/fimmu.2022.919100).
- The use of "PSP" is ambiguous. When referring to pneumococcal surface protein A (PspA), please use the correct terminology throughout the manuscript.
Reply.
We acknowledge the ambiguity in the phrase “A comparative assessment of the immunogenicity and protective efficacy of a recombinant tandem of immunogenic peptides from S. pneumoniae surface proteins (PSP) administered parenterally or orally was performed in a Balb/c mouse model.” To resolve this, we have revised the phrase in the introduction to: “A comparative assessment of the immunogenicity and protective efficacy of a recombinant tandem (PSP) of immunogenic peptides from S. pneumoniae surface proteins administered parenterally or orally was performed in a Balb/c mouse model.” In this study, we investigate the chimeric recombinant protein PSP, which is a sequential combination of immunogenic fragments from three S. pneumoniae surface proteins, including a fragment of PspA. Detailed information on protein production is described in our paper (doi: 10.4172/2157-7560.1000304).
- This study would benefit from a more comprehensive analysis of immune responses, including T cell responses and mucosal antibodies. This provided a more complete picture of the protective mechanisms involved.
Reply.
We agree with this observation. A comprehensive analysis of the mechanisms involved in the described phenomenon, including T cell responses and mucosal antibodies, is a priority for our future research..
- The current study only examined short-term protection. This study aimed to assess the durability of immune protection over a longer period of time.
Reply.
The aim of this study was to compare the protective efficacy of two vaccination methods. Indeed, the interval between the last vaccination and infection was approximately one month. To determine the true potential of probiotic vaccines within the broader framework of preventive measures, a thorough comparative analysis of the long-term effectiveness of the induced protection over extended periods is required. This is one of the key points we plan to address in future studies.
Minor Comments:
- In the Materials and Methods section, the subheaders for "Immunization and Evaluation of Immune Protection" and "Evaluation of Passive Immune Protection" are missing. Please add these details for an improved organization.
Reply.
It was done.
- The manuscript contains numerous typographical errors, missing spaces, and incorrect subscripts/superscripts. Please carefully proofread the manuscript.
Reply.
It was corrected.
- Standardize terminology throughout the study. For instance, choose either "Fetal calf serum" or "FBS" and use it consistently.
Reply.
It was done.
- Ensure that scientific names, such as Streptococcus pneumoniae, are spelled out in full at first mention and italicized appropriately.
Reply.
It was done.
- Provide a rationale for using the beta variant S1 fragment and consider discussing how results might differ with the current strains.
Reply.
A detailed discussion on the rationale for using the beta variant S1 fragment of the coronavirus, in light of the emergence of new antigenic variants, is presented in our article (doi: 10.3390/vaccines11111714) , which focuses on the development of a vaccine specific for SARS-CoV-2. In brief, the S1 fragment used in our study largely excludes the most variable RBD region of the spike protein, making it more conserved across variants. Since this article is not focused on addressing the prevention of coronavirus infection, but rather uses these data to illustrate the described immune response patterns in a viral infection model, we did not deem it necessary to discuss the efficacy of a specific vaccine formulation in detail.
Suggestions for Improvement.
Consider conducting dose optimization studies for both parenteral and oral administration to strengthen the comparative analysis.
Investigate the impact of different oral dosing schedules on immune responses.
Provide more mechanistic insights into how oral vaccination offers protection despite lower antibody titers.
Explore combinations of parenteral and oral vaccinations as a potential strategy to enhance overall protection.
Reply.
We are grateful for these valuable suggestions to further develop research related to the topic of this article. We will certainly take them into consideration in future studies. As a comment, we note that in 2024, we published an article addressing the use of a probiotic vaccine in a combined prime-boost immunization strategy (doi: 10.3390/ijms25010215).
Reviewer 2 Report
Comments and Suggestions for Authors
The study by Leontieva et al. designed a series of experiments to compare the effectiveness of vaccinations via parenteral or mucosal administration against SARS-CoV-2 and Streptococcus in two different in vivo animal models. Indeed, this is an interesting study with many main and minor results summarized in authors' manuscript. This reviewer can appreciate the authors' efforts trying to tell a brilliant story that "significant differences in the immunogenicity of parenteral and oral vaccine antigens, but comparable protective efficacy". Some major and minor points are listed below for authors to improve their manuscript.
1) The introduction section is too short and insufficient to tell a clear background and the significance of this study. Thus, the authors should rewrite this section to better present the mainstream and logic of this study.
2) The study lacks the confirmation and verification of the produced vaccine products with antigens, either via protein WB or immunostaining with proper antibodies, which is essential.
3) The main body of the study is too long and the authors should have considered shortening to one pathogen, either SARS-CoV2 or Streptococcus. Actually, there are differences between vaccines for pathogenic virus and bacteria in many aspects, e.g., vaccination dosage, adjuvants, efficacy, and immuno-response. It is not clear if those can be compared to draw any convincible conclusions.
3) Considering the relatively lower efficacy of mucosal vaccines using probiotic vaccines, the readers might have concerns that why the authors designed such experiments in this study. Long-term administration with same probiotic vaccines and continuous observations seem required in a case-by-case manner.
4) Histological analysis from this study Figure 6/11 are not very clear. What are the main conclusions?
5) All the Figure legends need significant improvement to better explain the design and results.
6) Reference needs update with more recent ones.
7) Language is also another main issue.
Comments on the Quality of English LanguageNeed major improvement.
Author Response
We greatly appreciate the reviewer’s insightful comments and the thorough review of our article. Their constructive and thoughtful feedback is highly valued.
1) The introduction section is too short and insufficient to provide a clear background and the significance of this study. The authors should rewrite this section to better present the rationale and logic of the study.
Reply.
We have revised the introduction to provide a clearer background, better present the rationale, and emphasize the significance of this study. The revised section now outlines the mainstream research and logic in a more comprehensive manner.
2) The study lacks confirmation and verification of the produced vaccine products with antigens, either via protein Western blotting or immunostaining with proper antibodies, which is essential.
Reply.
All vaccine products, including their generation, verification, and the evaluation of protective efficacy, have been thoroughly described in our previous publications. These are referenced in section 2.9 of the Materials and Methods.
3) The main body of the study is too long, and the authors should consider focusing on one pathogen, either SARS-CoV-2 or Streptococcus. There are differences between vaccines for viral and bacterial pathogens, such as dosage, adjuvants, efficacy, and immune responses. It is unclear if these can be compared to draw convincing conclusions.
Reply.
It was important for us to demonstrate that the patterns described in this study were observed in both viral and bacterial infections. For instance, in this experiment, the level of the humoral immune response, as measured by IgG antibody titers in serum, was not a correlate of protective efficacy for either the viral or bacterial infections. Therefore, we would prefer to keep both parts of the results in the article to emphasize the consistency of our findings across different pathogen types.
4) Considering the relatively lower efficacy of mucosal vaccines using probiotic vectors, readers might be concerned about why the authors designed such experiments. Long-term administration with the same probiotic vaccine and continuous observations seem required on a case-by-case basis.
Reply.
We demonstrated that although probiotic vaccines show relatively lower immunogenicity (based on humoral immune responses) compared to parenteral vaccines, their overall protective efficacy was comparable. The reviewer is correct that, in practical terms, long-term administration of probiotic vaccines might be needed to enhance or sustain the immune response over time. Unlike parenteral vaccines, probiotic oral vaccines, due to the beneficial properties of their bacterial vector, lack side effects and can be administered more frequently. Further research will be required to determine how such dosing regimens impact the immune response.
5) The histological analysis in Figures 6/11 is not very clear. What are the main conclusions?
Reply.
We have revised the text and added clarifications regarding the conclusions drawn from the histological analysis.
6) All the figure legends need significant improvement to better explain the experimental design and results.
Reply.
The figure legends have been revised for clarity and now better explain the design and outcomes of the experiments.
7) The references need to be updated with more recent sources.
Reply.
We have updated the references with more recent publications.
8) Language is another main issue.
Reply.
We have revised the manuscript to improve the language throughout.
Round 2
Reviewer 2 Report
Comments and Suggestions for Authors
This reviewer has no further major concerns on this revised version of manuscript.
Comments on the Quality of English Languageminor
Author Response
We greatly appreciate the reviewer’s insightful comments and the thorough review of our article. Their constructive and thoughtful feedback is highly valued.
The reviewer had no further comments in this round.